# SDF-CAR: 3D Coronary Artery Reconstruction from Two Views with a Hybrid SDF-Occupancy Implicit Representation

**Ahmed Reda**[1]                                                                    ES-AHMEDR.ALI2025@ALEXU.EDU.EG
**Mohamed Ashraf**[1]                                                         ES-MOHAMEDA.HAMDY2025@ALEXU.EDU.EG
**Mohamed G. Abdelkader**[1]                                              MOHAMED_ABDELKADER@ALEXU.EDU.EG
**Yasser Abuouf**[1]                                                              YASSER.ABUOUF@ALEXU.EDU.EG
**Muhamed Albadawi**[1]                                                        M.BADAWI@ALEXU.EDU.EG
[1] *Faculty of Engineering, Alexandria University*

**Editors:** Under Review for MIDL 2026

## Abstract

Three-dimensional (3D) reconstruction of coronary arteries is crucial for accurate diagnosis and treatment planning of cardiovascular diseases. Although Coronary Computed Tomography Angiography (CCTA) can generate 3D models, it involves high radiation doses, is costly, and cannot be used during real-time interventions. In contrast, Invasive Coronary Angiography (ICA), the standard imaging procedure during interventions, typically provides only a few 2D projections, making 3D geometry difficult to interpret. To bridge this gap, we propose a novel self-supervised framework that leverages a Signed Distance Field (SDF)-based neural implicit representation for reconstructing high-quality 3D coronary artery geometry from only two conventional 2D ICA projections. The proposed framework eliminates the need for 3D ground truth or large training datasets. Combining SDF-based geometric priors with an occupancy-based rendering loss, we achieve more stable optimization and higher-fidelity reconstructions than purely occupancy-based methods. Extensive experiments show that the proposed method outperforms the state-of-the-art baseline. It improves topological accuracy (cIDice) by over 16% and significantly reduces surface error. Source code is available at `https://github.com/reda1609/SDF-CAR`.

**Keywords:** 3D Reconstruction, Coronary Artery, Neural Implicit Representations, Signed Distance Fields, X-ray Angiography, Self-Supervised Learning, Sparse-View Reconstruction

## 1. Introduction

Cardiovascular diseases (CVDs) remain the leading cause of mortality worldwide (World Health Organization, 2021). X-ray invasive coronary angiography (ICA) is the clinical gold-standard for diagnosing and treating coronary artery disease, providing real-time guidance during interventions (Lashgari et al., 2024). However, ICA acquires only two-dimensional (2D) projections of the inherently three-dimensional (3D) coronary tree, obscuring true spatial geometry through vessel overlap and foreshortening, which complicates accurate assessment of lesion severity and treatment planning (Green et al., 2005).

Reconstructing 3D coronary geometry from ICA is thus clinically vital but challenging. To minimize radiation exposure, only two or three projections are typically acquired (Cimen et al., 2016), creating significant challenges: **extreme vessel sparsity**, **non-rigid motion**

between non-simultaneous projections, and critically, **complete absence of 3D ground truth** for real ICA.

While supervised methods (Wang et al., 2024c) show strong performance, they require unavailable paired 2D-3D ground truth data. This has driven interest in self-supervised approaches like **NeCA** (Wang et al., 2024b), which uses an **occupancy network** for patient-specific optimization. However, occupancy networks lack **surface-awareness**, providing weak geometric priors that often yield "blobby" reconstructions with broken vessel connectivity.

We posit that **Signed Distance Functions (SDFs)** provide the missing geometric prior. SDFs encode continuous surface information by storing distances to the nearest surface, inherently encouraging smoothness and connectivity (see Appendix A). However, pure SDF representations can be unstable with only two views.

We introduce a novel framework that **synergistically combines SDF representation with occupancy-based rendering**. Instead of replacing occupancy, we use the SDF as an internal geometric prior while retaining a stable occupancy renderer. This hybrid approach leverages both the robustness of occupancy probability and the geometric precision of SDFs.

Our main contributions are summarized as follows:

1. We propose **SDF-CAR**, a self-supervised framework that introduces a **combined occupancy and SDF representation** for 3D coronary reconstruction. To our knowledge, this is the first work to leverage SDF-based geometric priors to enhance an occupancy-network optimization in this domain.

2. We demonstrate that our **combined loss function**, which uses the SDF to constrain the occupancy optimization, provides superior geometric priors for tubular structures compared to a pure occupancy network, leading to enhanced topological accuracy and surface quality.

3. We perform an extensive evaluation on public CCTA datasets, showing that SDF-CAR achieves state-of-the-art performance, outperforming leading self-supervised and supervised baselines.

## 2. Related Works

### 2.1. Traditional Methods for 3D Coronary Reconstruction

Early approaches like epipolar geometry (Banerjee et al., 2019b), NURBS (Galassi et al., 2018), and point-cloud reconstruction (Banerjee et al., 2020) relied heavily on geometric principles. These techniques generally require laborious manual annotation of vessel centerlines and correspondence matching. Furthermore, their inability to compensate for non-rigid cardiac and respiratory motion between non-simultaneous projections (Banerjee et al., 2019a) significantly limits their clinical utility and automation potential.

### 2.2. Deep Learning for 3D Medical Image Reconstruction

Deep learning has advanced sparse-view CT reconstruction (Wang et al., 2024a, 2020b; Zha et al., 2022), though these methods typically require dozens of projections—infeasible

for ICA. For bi-planar X-ray, models like X2CT-GAN (Ying et al., 2019) and CCX-RayNet (Ratul et al., 2021) handle simultaneous projections but do not address the non-rigid motion inherent in sequential ICA scans. Our work distinguishes itself by targeting the extreme sparse-view case (two projections) under complex motion conditions.

## 2.3. Deep Learning-based 3D Coronary Reconstruction

Recent deep learning approaches are broadly divided into supervised and self-supervised paradigms.

**Supervised Methods** like **DeepCA** (Wang et al., 2024c) employ GANs trained on simulated projections (Iyer et al., 2023; Wang et al., 2020a) to predict 3D structure. While effective, they depend on large, paired 3D ground truth datasets unavailable for real ICA and may struggle with domain gaps between synthetic training data and real clinical acquisitions.

**Self-Supervised Methods** optimize patient-specific models using only 2D projections. **NeCA** (Wang et al., 2024b) pioneered this using a neural implicit **occupancy network**. However, occupancy networks lack explicit surface definition, often yielding "blobby" reconstructions with broken connectivity. Another approach, **3DGR-CAR** (Fu et al., 2024), uses 3D Gaussians but can be memory-intensive for high-resolution volumes. In contrast, our SDF representation offers a continuous, memory-efficient alternative with superior geometric priors.

## 2.4. Neural Implicit Representations and SDFs

Neural Implicit Representations (INRs) map coordinates to signals like occupancy (Mescheder et al., 2019) or signed distance (Park et al., 2019), popularized by NeRF (Mildenhall et al., 2020; Müller et al., 2022; Barron et al., 2021). While **occupancy networks** lack surface information, **Signed Distance Functions (SDFs)** explicitly represent surfaces as zero-level sets (Yariv et al., 2021). We posit that the SDF's inductive bias for smoothness and continuity makes it ideally suited for reconstructing the intricate, tubular geometry of coronary arteries.

## 3. Method

Our goal is to reconstruct a 3D model of the coronary artery tree, represented as a neural implicit function, from only two 2D X-ray projections. Our framework, SDF-CAR, builds upon the self-supervised paradigm of NeCA (Wang et al., 2024b) but introduces a critical enhancement: a hybrid representation that leverages the geometric precision of a Signed Distance Function (SDF) to guide the optimization of an occupancy-based renderer.

## 3.1. Problem Formulation

The input to our model is a set of $N$ (where $N = 2$) 2D X-ray projections $\{I_i\}_{i=1}^{N}$ with known camera geometries $\{\mathbf{\Pi}_i\}_{i=1}^{N}$. Following NeCA (Wang et al., 2024b), we represent the 3D volume using a neural implicit function. The core of the method is a multi-layer perceptron (MLP) $f$ that maps a 3D coordinate $\mathbf{x} = (x, y, z)$ to a value representing the scene.

In NeCA, this network is an **occupancy network**, trained to predict a probability $p \in [0, 1]$ using a sigmoid activation. In this work, we modify the network to predict an **unbounded continuous scalar field** $f_{\text{field}}(\mathbf{x}) = s \in \mathbb{R}$. While we initialize this field to approximate a Signed Distance Function (SDF), we do not strictly enforce the Eikonal constraint ($|\nabla f| = 1$) during training. Instead, we leverage the unbounded nature of this representation to provide stronger gradients and sharper surface definitions than sigmoid-saturated occupancy networks. The zero-level set $\{\mathbf{x}|f_{\text{field}}(\mathbf{x}) = 0\}$ implicitly defines the vessel surface.

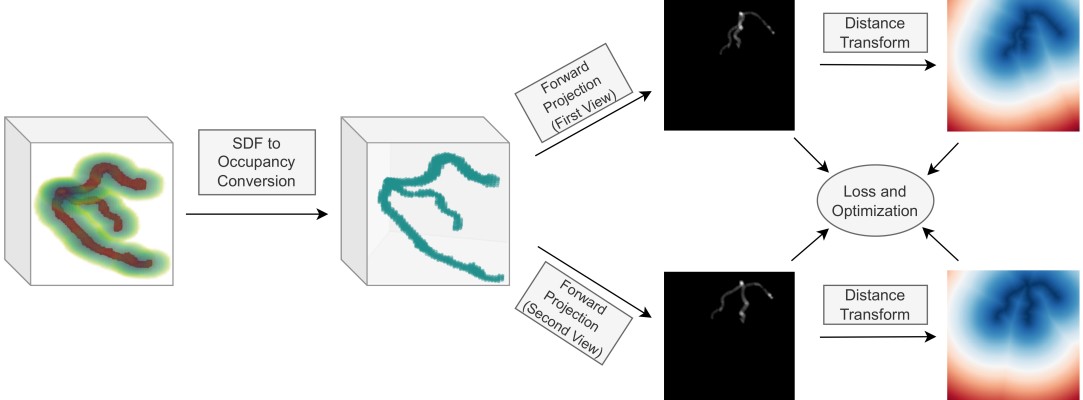

Figure 1: Overview of our SDF-CAR framework. The pipeline shows the conversion from 3D Signed Distance Function (SDF) representation to 2D projections. The 3D SDF (left) is converted to an occupancy volume using a scaled sigmoid function, which is then projected to generate 2D occupancy projections. Simultaneously, the SDF values are used to compute 2D SDF projections through distance transform operations. Both projection types are used in our combined loss function during optimization.

## 3.2. SDF-CAR Framework

Our model consists of three core stages: (1) coordinate encoding and SDF prediction, (2) a hybrid SDF-to-occupancy conversion for differentiable rendering, and (3) a combined loss function for optimization.

### 3.2.1. COORDINATE ENCODING AND SDF PREDICTION

For a given 3D coordinate $\mathbf{x}$, we first normalize it to the volume space. We then employ the efficient multiresolution hash encoding $H$ from Instant-NGP (Müller et al., 2022) to project the coordinate into a higher-dimensional feature vector $\mathbf{h} = H(\mathbf{x})$. This feature vector is passed through an 8-layer residual MLP $f_\theta$ to predict the signed distance:

$$s = f_\theta(\mathbf{h}),$$

where $\theta$ represents the trainable parameters of the MLP. This constitutes our SDF representation of the coronary tree.

### 3.2.2. DIFFERENTIABLE RENDERING VIA SDF-TO-OCCUPANCY CONVERSION

To render a 2D projection from our 3D SDF, we require a differentiable function that relates the SDF to X-ray attenuation. A direct physical model is complex and unstable for optimization. Instead, we leverage a probabilistic conversion, noting that points near the surface (where $s \approx 0$) are most likely to contribute to attenuation.

As illustrated in Figure 1, we convert the predicted SDF value $s$ at each point to an occupancy probability $o$ using a scaled sigmoid function:

$$o = \sigma(-\alpha \cdot s) = \frac{1}{1 + e^{\alpha s}},$$

where $\alpha > 0$ is a sharpness parameter. This function provides a smooth, differentiable transition from 0 (definitely outside) to 1 (definitely inside), with the transition region becoming narrower as $\alpha$ increases. This creates a dense 3D occupancy volume $V_{\mathrm{occ}}$ from our SDF.

Finally, we use the same differentiable cone-beam forward projector $\mathcal{P}$ as in NeCA (Wang et al., 2024b) to render a 2D occupancy projection $\hat{I}_{\mathrm{occ}}$ from the volume $V_{\mathrm{occ}}$, simulating the X-ray attenuation process:

$$\hat{I}_{\mathrm{occ}} = \mathcal{P}(V_{\mathrm{occ}}; \mathbf{\Pi}).$$

### 3.2.3. COMBINED GEOMETRIC AND OCCUPANCY LOSS

A key innovation of SDF-CAR is the use of a multi-objective loss function. For each input projection $I_i$, we generate a binary segmentation mask $M_i$. From this mask, we compute two ground-truth targets:

1. **2D Occupancy Target:** The binary mask itself, $G_{\mathrm{occ}} = M_i$.

2. **2D Distance Transform Target:** We apply a 2D Euclidean distance transform to $M_i$ to compute a continuous Distance Map $G_{\mathrm{dt}}$.

Our loss function is a weighted combination of the occupancy error and the geometric distance error:

$$\mathcal{L}_{\mathrm{total}} = \lambda_1 \cdot \mathcal{L}_{\mathrm{geo}} + \lambda_2 \cdot \mathcal{L}_{\mathrm{occ}} \tag{1}$$

$$\mathcal{L}_{\mathrm{geo}} = \frac{1}{|\Omega|} \sum_{p \in \Omega} \| \mathrm{DT}(M_i(p)) - \mathrm{DT}(\hat{I}_{\mathrm{occ}}(p)) \|_2^2 \tag{2}$$

$$\mathcal{L}_{\mathrm{occ}} = \frac{1}{|\Omega|} \sum_{p \in \Omega} \| M_i(p) - \hat{I}_{\mathrm{occ}}(p) \|_2^2 \tag{3}$$

where $\Omega$ is the image domain, $\mathrm{DT}(\cdot)$ denotes the 2D distance transform operation, and $\mathcal{L}_{\mathrm{geo}}$ serves as our geometric supervision term. To ensure end-to-end differentiability of the geometric term $\mathcal{L}_{\mathrm{geo}}$, we utilize the differentiable distance transform implementation

provided by the Kornia library (Eriba et al., 2019), which approximates the Euclidean distance field via a differentiable convolution-based propagation. This combination allows the model to learn topological connectivity via the unbounded distance field while maintaining projection consistency via the occupancy renderer.

## 4. Experiments

We evaluated SDF-CAR on the publicly available ImageCAS dataset (Zeng et al., 2023), comparing against the state-of-the-art self-supervised baseline NeCA (Wang et al., 2024b). Following the experimental protocol of NeCA, we conducted separate evaluations on the Right Coronary Artery (RCA) and Left Anterior Descending (LAD) artery. Implementation details and evaluation metrics are provided in Appendix B.

### 4.1. Quantitative and Qualitative Results

#### 4.1.1. Right Coronary Artery (RCA) Evaluation

**Quantitative Results:** Table 1 presents the numerical comparison on the RCA dataset. SDF-CAR demonstrates substantial improvements over the baseline, increasing the topological **cIDice score by over 16 percentage points (91% vs 74.29%)** and reducing the surface error $CD_{\ell_2}$ **by more than half (0.51mm vs 1.12mm)**. These metrics indicate a dramatic reduction in topological breaks and surface irregularities.

Table 1: Quantitative comparison on Right Coronary Artery (RCA) dataset. Best results are in **bold**.

| Method | cIDice (%) | Dice (%) | IoU (%) | reError | $CD_{\ell_2}$ (mm) | reMSE ($\times 10^{-4}$) |
|---|---|---|---|---|---|---|
| NeCA | $74.29 \pm 15.25$ | $75.01 \pm 15.61$ | $62 \pm 16.45$ | $0.4 \pm 0.99$ | $1.12 \pm 2.46$ | $2.2 \pm 1.5$ |
| SDF-CAR | $\mathbf{91 \pm 4.5}$ | $\mathbf{91.6 \pm 3.6}$ | $\mathbf{84.7 \pm 6}$ | $\mathbf{0.12 \pm 0.05}$ | $\mathbf{0.51 \pm 0.38}$ | $\mathbf{1.02 \pm 0.94}$ |

**Qualitative Results:** The visual comparison in Figure 2 corroborates the quantitative findings. As observed in the baseline reconstructions, the pure occupancy representation often fails to maintain connectivity in curved segments, resulting in broken vessels and "blobby" artifacts. In contrast, SDF-CAR successfully reconstructs the continuous tubular structure of the RCA, preserving smoothness even in regions with complex curvature.

#### 4.1.2. Left Anterior Descending (LAD) Evaluation

**Quantitative Results:** For the LAD dataset, which often involves finer vessel branches, our method continues to outperform the baseline. As shown in Table 2, SDF-CAR achieves a **cIDice of 90.52%** compared to the baseline's 77.17%. This demonstrates that our hybrid representation is robust across different anatomical variations and vessel calibers.

**Qualitative Results:** Figure 3 illustrates the reconstruction quality on the LAD anatomy. The baseline method frequently loses high-frequency details, causing thin distal branches to vanish or become disconnected. SDF-CAR leverages the signed distance prior to maintain the structural integrity of these thinner vessels, resulting in a reconstruction that is topologically faithful to the ground truth.

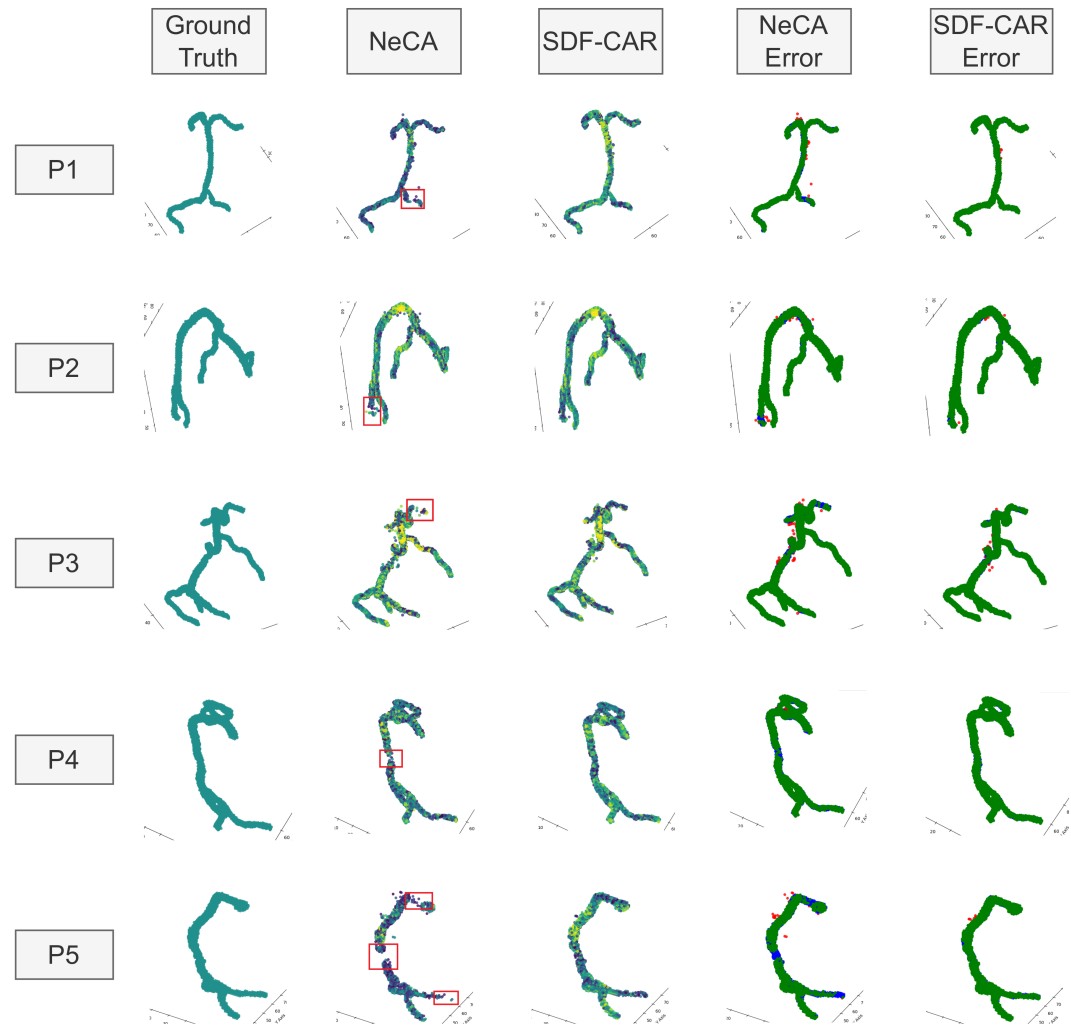

Figure 2: Qualitative comparison of Right Coronary Artery (RCA) reconstruction across five patients (rows). The columns display: (1) Ground Truth (GT), (2) NeCA raw output, (3) Our SDF-CAR raw output, (4) NeCA error overlay, and (5) SDF-CAR error overlay. In the overlay columns, **Green** points represent correct overlap, **Red** points represent predicted deviations, and **Blue** points represent missed ground truth structures. Red rectangles highlight specific areas of topological disconnection in the baseline that are successfully resolved by our method.

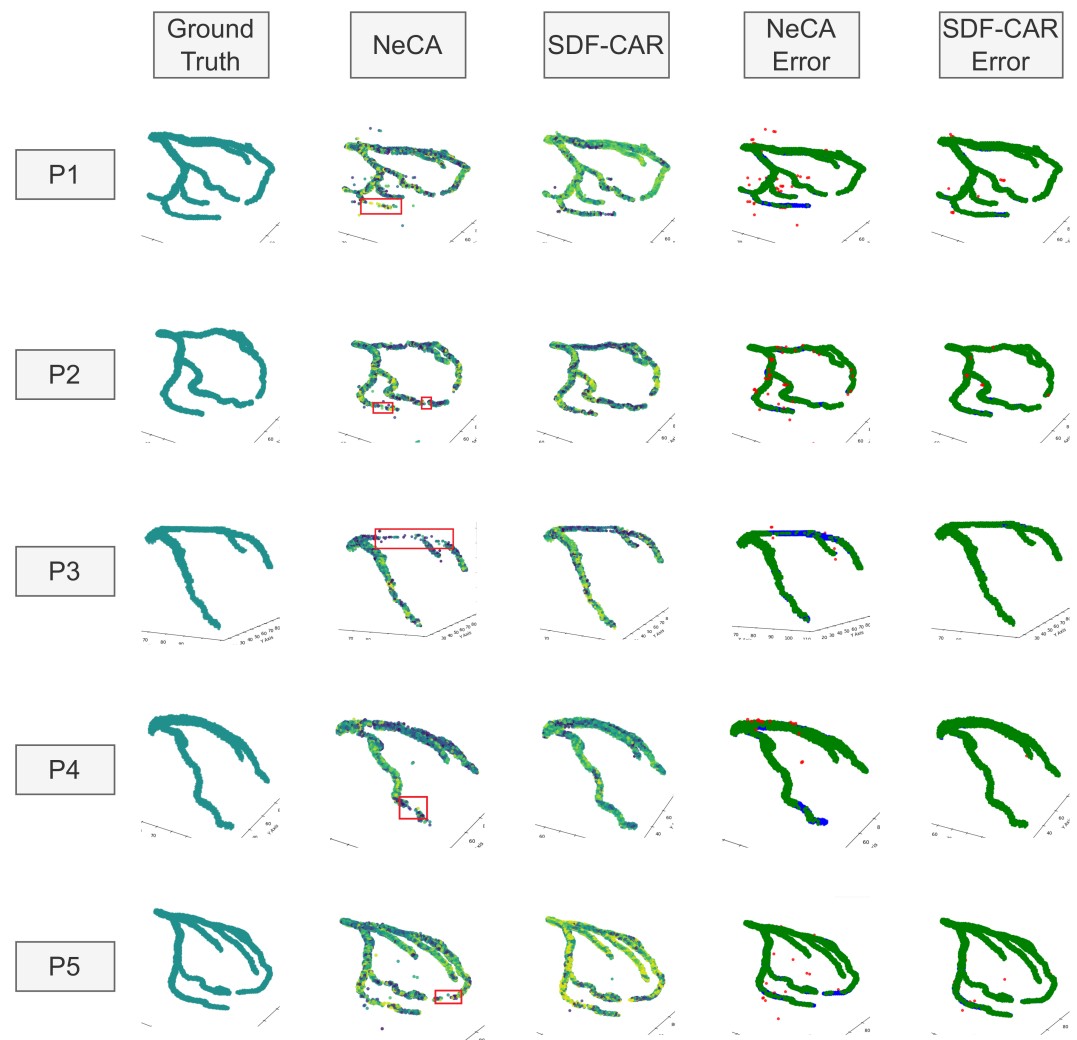

Figure 3: Qualitative comparison of Left Anterior Descending (LAD) artery reconstruction across five patients (rows). Columns correspond to: (1) GT, (2) NeCA raw output, (3) SDF-CAR raw output, (4) NeCA overlay comparison, and (5) SDF-CAR overlay comparison. The color coding follows: **Green** (Correct Overlap), **Red** (Predicted only), and **Blue** (GT only/Missed). Red rectangles highlight specific regions where the baseline method fails to reconstruct thin distal branches or creates disconnected segments, artifacts that are successfully resolved by our SDF-CAR method.

Table 2: Quantitative comparison on Left Anterior Descending (LAD) dataset. Best results are in **bold**.

| Method | cIDice (%) | Dice (%) | IoU (%) | reError | $CD_{\ell_2}$ (mm) | reMSE ($\times 10^{-4}$) |
|---|---|---|---|---|---|---|
| NeCA | $77.17 \pm 7.8$ | $80.85 \pm 7.2$ | $68.39 \pm 9.16$ | $0.22 \pm 0.06$ | $0.92 \pm 0.43$ | $2.33 \pm 1.48$ |
| SDF-CAR | $\mathbf{90.52 \pm 5.46}$ | $\mathbf{92.72 \pm 4.2}$ | $\mathbf{86.7 \pm 6.97}$ | $\mathbf{0.1 \pm 0.05}$ | $\mathbf{0.6 \pm 0.43}$ | $\mathbf{1.03 \pm 0.86}$ |

## 4.2. Ablation Study

To disentangle the contributions of the proposed hybrid SDF representation and the combined loss function, we conducted a comprehensive ablation study comparing three configurations. We present them in increasing order of complexity and performance:

1. **Occ. Rep. + Combined Loss:** A standard occupancy network (sigmoid output) optimized using our combined loss (Eq. 1). This isolates the effect of the loss function.

2. **SDF Rep. + Occ. Loss:** Our SDF-based network optimized using only the occupancy loss (Eq. 3).

3. **SDF-CAR (Ours):** The full framework combining the SDF representation with the combined loss.

Table 3 and Table 4 present the results for RCA and LAD datasets, respectively.

Table 3: Ablation study on Right Coronary Artery (RCA) dataset. Ordered by performance improvement. Best results are in **bold**.

| Configuration | cIDice (%) | Dice (%) | IoU (%) | reError | $CD_{\ell_2}$ (mm) | reMSE ($\times 10^{-4}$) |
|---|---|---|---|---|---|---|
| Occ. Rep. + Combined Loss | $76.29 \pm 10.68$ | $77.05 \pm 10.77$ | $63.81 \pm 13.26$ | $0.27 \pm 0.10$ | $0.77 \pm 0.34$ | $2.18 \pm 1.51$ |
| SDF Rep. + Occ. Loss | $90.56 \pm 6.4$ | $91.11 \pm 4.84$ | $84.01 \pm 7.7$ | $0.13 \pm 0.06$ | $0.52 \pm 0.38$ | $1.07 \pm 0.1$ |
| **SDF-CAR (Ours)** | $\mathbf{91 \pm 4.5}$ | $\mathbf{91.6 \pm 3.6}$ | $\mathbf{84.7 \pm 6}$ | $\mathbf{0.12 \pm 0.05}$ | $\mathbf{0.51 \pm 0.38}$ | $\mathbf{1.02 \pm 0.94}$ |

Table 4: Ablation study on Left Anterior Descending (LAD) artery dataset. Best results are in **bold**.

| Configuration | cIDice (%) | Dice (%) | IoU (%) | reError | $CD_{\ell_2}$ (mm) | reMSE ($\times 10^{-4}$) |
|---|---|---|---|---|---|---|
| Occ. Rep. + Combined Loss | $77.51 \pm 7.79$ | $81.15 \pm 6.75$ | $68.77 \pm 8.80$ | $0.22 \pm 0.07$ | $0.91 \pm 0.43$ | $2.25 \pm 1.41$ |
| SDF Rep. + Occ. Loss | $89.07 \pm 8.6$ | $91.79 \pm 6.3$ | $85.37 \pm 9.58$ | $0.13 \pm 0.16$ | $\mathbf{0.82 \pm 0.96}$ | $1.35 \pm 1.54$ |
| **SDF-CAR (Ours)** | $\mathbf{90.52 \pm 5.46}$ | $\mathbf{92.72 \pm 4.2}$ | $\mathbf{86.7 \pm 6.97}$ | $\mathbf{0.1 \pm 0.05}$ | $\mathbf{0.6 \pm 0.43}$ | $\mathbf{1.03 \pm 0.86}$ |

The results clearly highlight the source of our method's performance. As shown in the first row of Table 3, the *Occ. Rep. + Combined Loss* configuration yields the lowest scores (e.g., RCA cIDice of 76.29%), confirming that the distance-transform loss alone cannot compensate for the limitations of a pure occupancy representation.

Introducing the Unbounded Field representation (Second Row) provides a massive jump in performance, raising the RCA cIDice to 90.56%. This finding is significant as it suggests

that the **unbounded nature of the implicit field** allows for superior gradient flow and surface definition compared to probability-bounded networks, even without enforcing strict Eikonal regularization. Finally, the full **SDF-CAR** framework (Third Row) achieves the best metrics across the board, demonstrating that the synergistic combination of an unbounded geometric representation and our hybrid loss is optimal.

## 5. Discussion and Limitations

While SDF-CAR demonstrates state-of-the-art performance, several limitations remain. First, validation on the ImageCAS dataset with clean CCTA scans does not capture real clinical ICA challenges such as cardiac motion, noise, and overlapping structures. Future work will integrate motion compensation modules to handle non-rigid deformations in clinical data.

Second, patient-specific optimization takes approximately 80 minutes on a T4 GPU, precluding real-time application during catheterization. Future research will explore meta-learning or encoder-based initialization to accelerate convergence. Finally, extremely thin side branches obscured in one view may still be difficult to reconstruct; temporal information from angiographic video sequences could potentially resolve these ambiguities.

## 6. Conclusion

We introduced SDF-CAR, a self-supervised framework for 3D coronary artery reconstruction from two X-ray projections. By combining a Signed Distance Function representation with occupancy-based rendering, we overcome the topological limitations of pure occupancy networks, achieving superior vessel connectivity and surface accuracy. Extensive experiments confirmed that SDF-CAR achieves state-of-the-art performance, demonstrating that incorporating explicit geometric knowledge into neural implicit representations is a powerful direction for medical image reconstruction.

## Acknowledgments

We thank the open-source community for the NeCA and Instant-NGP repositories which facilitated this research.

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

# Appendix A. Projection Representations

The choice of representation fundamentally influences the optimization landscape and reconstruction quality. Traditional occupancy-based representations encode space as a binary classification problem: a point is either inside (1) or outside (0) the object. While this provides a straightforward formulation, it lacks explicit geometric information about surface proximity and orientation, particularly critical for thin tubular structures like coronary arteries.

In contrast, Signed Distance Functions (SDFs) encode the signed distance from any point in space to the nearest surface, with negative values inside the object and positive values outside. This continuous representation inherently captures surface geometry and provides smoother gradients during optimization. As illustrated in Figure 4, when projected to 2D, occupancy representations yield binary masks that only distinguish vessel from background. SDF projections, however, preserve rich geometric information through distance fields, where pixel intensities encode proximity to vessel boundaries. This distance information serves as a powerful geometric prior, encouraging the optimization to favor smooth, connected surfaces over fragmented or bloated reconstructions.

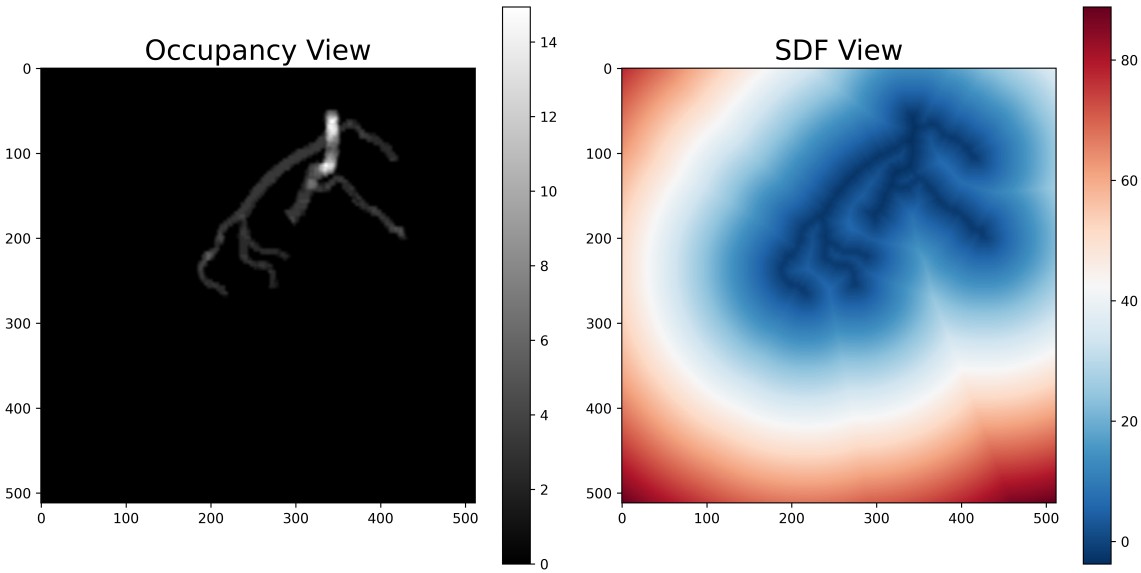

Figure 4: Visual comparison of 2D projection representations for a coronary artery. **Left:** Standard binary **Occupancy** projection, where voxels are simply labeled as vessel (white) or background (black). **Right: Signed Distance Function (SDF)** projection, where each pixel's value represents the distance to the nearest vessel surface.

## Appendix B. Experimental Setup

### B.1. Datasets

We evaluated our proposed SDF-CAR framework on the publicly available ImageCAS dataset (Zeng et al., 2023), which contains 3D coronary artery trees segmented from Coronary Computed Tomography Angiography (CCTA) scans. Following the experimental protocol of NeCA (Wang et al., 2024b), we conducted separate evaluations on the Right Coronary Artery (RCA) and Left Anterior Descending (LAD) artery to assess the generalizability of our method across different vessel anatomies. The dataset consisted of 68 samples each for RCA and LAD evaluation.

### B.2. Implementation Details

Our SDF-CAR framework was implemented using PyTorch and optimized with the Adam optimizer (Kingma and Ba, 2014) using a fixed learning rate of $1 \times 10^{-4}$. Each patient-specific optimization was conducted for 5000 epochs to ensure convergence. All experiments were performed on NVIDIA T4 GPU. The multiresolution hash encoder was configured with 16 resolution levels, a hash table size of $2^{19}$, and 2 feature dimensions per entry, following the Instant-NGP architecture (Müller et al., 2022).

For the SDF-to-occupancy conversion, the sharpness parameter was set to $\alpha = 50$. We empirically found that this value provided the optimal balance between gradient flow and surface sharpness; lower values ($\alpha < 30$) resulted in overly blurry boundaries, while higher values ($\alpha > 80$) led to unstable gradients during optimization. The loss weights were empirically set to $\lambda_1 = 1.00$ and $\lambda_2 = 0.25$ for the SDF and occupancy losses, respectively.

### B.3. Evaluation Metrics

To comprehensively evaluate the reconstruction quality, we employed six metrics that assess different aspects of the reconstruction:

- **Topology Preservation:** Centerline Dice score (cIDice) (Shit et al., 2021)

- **Segmentation Accuracy:** Dice Similarity Coefficient (Dice) and Intersection over Union (IoU)

- **Surface Quality:** Chamfer $\ell_2$ Distance ($\mathrm{CD}_{\ell_2}$)

- **Reconstruction Error:** Reconstruction Error (reError) and Reconstruction Mean Squared Error (reMSE)

