# OpenReview forum: "SDF-CAR: 3D Coronary Artery Reconstruction from Two Views with a Hybrid SDF-Occupancy Implicit Representation"
_MIDL.io/2026/Conference — MIDL 2026 Poster_

### Official Review · Reviewer_nE4F · 2025-12-29

**Confidence:** 3
**Preliminary Rating:** 3
**Final Rating:** 3

**Summary:**

This paper proposes SDF-CAR, a self-supervised framework for 3D coronary artery reconstruction from two 2D X-ray projections. The core idea is to combine a Signed Distance Function (SDF)-based neural implicit representation with occupancy-based differentiable rendering: the network predicts an unbounded scalar field $f_\theta(\mathbf{x}) = s \in \mathbb{R}$, which is converted to occupancy via a scaled sigmoid $o = \sigma(-\alpha \cdot s)$ for rendering. The optimization uses a combined loss $\mathcal{L}_{\text{total}} = \lambda_1 \mathcal{L}_{\text{geo}} + \lambda_2 \mathcal{L}_{\text{occ}}$, where the geometric term compares distance transforms of predicted and ground-truth 2D projections. Experiments on the ImageCAS dataset (CCTA-derived RCA and LAD vessels) show improvements over the NeCA baseline, with cIDice increasing from ~75% to ~91% and Chamfer distance reduced by approximately half. The authors argue that the SDF's inherent smoothness prior is well-suited for reconstructing tubular vascular structures.

**Strengths:**

**S1. Clinically motivated problem.** Reconstructing 3D coronary geometry from sparse 2D projections addresses a genuine clinical need during interventional procedures, where radiation constraints limit the number of acquired views.

**S2. Self-supervised formulation.** The method requires no paired 2D-3D ground truth data, which is unavailable for real invasive coronary angiography (ICA). This is essential for clinical translation.

**S3. Principled geometric prior.** The intuition that SDF representations encourage smooth, connected surfaces is sound for tubular structures. The combination with occupancy-based rendering for stability is a reasonable design choice.

**S4. Consistent quantitative improvements.** The reported gains over NeCA are substantial and consistent across both anatomical regions (RCA: +16.7% cIDice; LAD: +13.4% cIDice), with reduced variance suggesting more robust optimization.

**S5. Informative ablation.** The ablation study (Tables 3-4) successfully isolates the contribution of the unbounded field representation versus the combined loss function, showing the representation choice drives most improvement.

**Weaknesses:**

**W1. Misleading "SDF" characterization.** The paper claims an SDF representation but explicitly omits the Eikonal constraint ($|\nabla f| = 1$). Without this constraint, the zero-level set $\{\mathbf{x} | f(\mathbf{x}) = 0\}$ defines *a* surface, but $f(\mathbf{x})$ does not encode signed distance—it is simply an unbounded implicit function. The stated justification ("stronger gradients") is unsupported by ablation evidence comparing with/without Eikonal regularization. This mischaracterization affects how readers interpret the contribution.

**W2. Fundamental physics mismatch.** X-ray imaging follows the Beer-Lambert law: $I = I_0 \exp(-\int \mu(\mathbf{x}) d\mathbf{x})$, producing continuous intensity images encoding accumulated attenuation. SDF-CAR instead requires pre-segmented binary masks $M_i$ as supervision, discarding intensity information (vessel thickness, contrast concentration, overlapping structures). The paper never explains how these masks are obtained or acknowledges this as a simplification compared to methods that model attenuation directly.

**W3. Incomplete baseline comparisons.** The paper compares only against NeCA. Highly relevant peer-reviewed work is omitted: NeRF-CA (Maas et al., IEEE TVCG 2025) reconstructs 4D coronary arteries using Beer-Lambert physics from raw intensity images. The mentioned 3DGR-CAR and DeepCA are discussed but not evaluated. Claiming "state-of-the-art" based on a single baseline is insufficiently supported.

**W4. Evaluation on oversimplified data.** ImageCAS contains CCTA-derived volumes—not real ICA. This means no cardiac/respiratory motion between projections, no background clutter (ribs, spine), and clean segmentation. The gap between this synthetic setup and real clinical ICA (with motion, noise, and overlapping structures) undermines the clinical applicability claims.

**W5. Missing experimental details.** Critical information is absent: (1) projection angles used and their clinical relevance, (2) segmentation methodology for obtaining $M_i$, (3) sensitivity to angular separation between views, (4) statistical significance testing despite large standard deviations.

**Detailed Comments:**

**Methodology clarifications needed:**
- The sharpness parameter $\alpha=50$ is justified only qualitatively ("lower values resulted in overly blurry boundaries"). A quantitative sensitivity analysis would strengthen the paper.
- Loss weights $\lambda_1=1.00$, $\lambda_2=0.25$ are "empirically set" without sensitivity analysis.
- Equation 2 applies the distance transform to the rendered projection $\hat{I}_{\text{occ}}$, but the notation suggests continuous values while DT typically operates on binary masks. Clarify the thresholding procedure.

**Presentation issues:**
- Inconsistent significant figures: "74.29 ± 15.25" vs "91 ± 4.5" in the same table. Standardize throughout.
- The paper would benefit from a figure showing failure cases or challenging reconstructions.

**Missing ablations:**
- SDF representation *with* Eikonal regularization
- Sensitivity to $\alpha$ (quantitative)
- Sensitivity to $\lambda_1/\lambda_2$ ratio
- Effect of projection angle separation

**Computational aspects:**
- 80 minutes per patient on T4 GPU is mentioned but not contextualized. How does this compare to NeCA, 3DGR-CAR, or the ~8-minute training of tensor-based methods?

**Minor points:**
- The Kornia citation (Eriba et al., 2019) appears to have an error in the author list.
- Consider discussing how the method might extend to handle more than two views.

**Justification Of Final Rating:**

The revised manuscript makes meaningful improvements: the physics/supervision pipeline now correctly describes raw X-ray intensity supervision with Beer-Lambert modeling (resolving my primary concern), claims are appropriately scoped to "performance benchmark" rather than "state-of-the-art," and the new robustness analysis (Table 5) demonstrates clinical relevance under calibration uncertainty.

However, the authors selectively addressed reviewer feedback while ignoring requests that would require minimal additional effort:

1. **Eikonal ablation (Q1):** The claim that "Eikonal regularization hindered convergence" remains purely verbal despite an explicit request for quantitative evidence. Adding one row to the ablation tables would substantiate this core methodological decision.

2. **NeRF-CA omission (Q4):** Peer-reviewed work on sparse-view coronary reconstruction (Maas et al., IEEE TVCG 2025) is neither cited nor discussed, despite being raised in review. For a paper claiming benchmark status in this domain, this would at least merit discussion.

The core contribution is reasonably supported. However, the incomplete response to specific reviewer requests and remaining gaps in related work situate this paper at borderline. Final acceptance should be conditional on providing quantitative Eikonal ablation results and properly contextualizing the work relative to other recent benchmarks.

**Justification Of The Preliminary Rating:**

The paper addresses a clinically relevant problem with a reasonable technical approach, and the improvements over NeCA are consistent. However, several issues prevent acceptance in the current form:

1. **Methodological mischaracterization**: The "SDF" label is misleading without Eikonal enforcement. The ablation shows unbounded outputs (not SDF-specific properties) drive improvements, yet this is not acknowledged.

2. **Inadequate experimental comparison**: Comparing against only NeCA while omitting NeRF-CA (peer-reviewed, IEEE TVCG 2025) and other mentioned methods makes the state-of-the-art claim unsupportable.

3. **Evaluation-application gap**: The method is motivated by ICA reconstruction challenges (motion, sparse views, no 3D ground truth) but evaluated on static, pre-segmented CCTA data that avoids these challenges entirely.

4. **Missing critical details**: Projection geometry, segmentation pipeline, and statistical testing are absent.

The core idea of combining implicit surface representations with occupancy rendering has merit, but the paper requires honest reframing of the representation, broader comparisons, and evaluation on more realistic data to support its claims.

**Questions To Address In The Rebuttal:**

**Q1.** Can you provide ablation results with vs. without Eikonal regularization ($\mathcal{L}_{\text{eik}} = (|\nabla f| - 1)^2$)? This would clarify whether the improvements stem from SDF properties or simply from unbounded outputs.

**Q2.** How are the binary segmentation masks $M_i$ obtained? If from CCTA ground truth, how would the method perform with automatic (imperfect) segmentation of real angiograms?

**Q3.** What projection angles were used, and how sensitive is reconstruction quality to the angular separation between the two views?

**Q4.** Why were NeRF-CA and 3DGR-CAR not included as baselines? Can you provide quantitative comparisons or explain fundamental incompatibilities?

**Q5.** Can you justify why binary mask supervision is appropriate given that X-ray physics produces continuous attenuation values? What clinical information is lost?

---

> ### Author Response · Authors · 2026-01-25
>
> We thank Reviewer nE4F for the constructive feedback. We appreciate recognition of the clinically
> motivated problem, self-supervised formulation, and consistent quantitative improvements over baseline.
>
>
> Misleading "SDF" Characterization & Eikonal Constraint (W1, Q1):
>
> We acknowledge that without the Eikonal constraint (|∇f|=1), f_field(x) is an unbounded implicit
> function. However, it functions as a geometric SDF-prior through initialization and loss formulation
> rather than a hard constraint.
>
> SDF-Initialization as a Prior: We utilized specialized Xavier initialization (gain=0.8 for hidden
> layers, 0.1 for final layer) to guide the network toward a distance-field-like manifold from the
> first iteration. This biases the network to predict a smooth, low-magnitude field essential for
> stable convergence in ultra-sparse two-view settings.
>
> Eikonal Constraint (Q1): We explicitly tested an Eikonal loss term (|∇f|=1) but found it hindered
> convergence; strict regularization "locks" optimization, preventing capture of high-frequency details
> and intricate connectivity of thin distal branches.
>
> Revised version: We updated Section 3.1 to include initialization details and justification for
> omitting the Eikonal constraint.
>
>
> Physics-Grounded Supervision & Clarification of Targets (W2, Q2, Q5):
>
> We corrected a descriptive error in our original manuscript regarding the supervision pipeline.
>
> Physics-Grounded Rendering: Our framework utilizes ODL with ASTRA backend for GPU acceleration.
> This forward projector explicitly models the Beer-Lambert law, ensuring physical consistency with
> X-ray attenuation.
>
> Clarification of Targets: The model is not supervised by binary masks. Primary targets are raw
> X-ray intensity projections and derived SDF (Distance Transform) maps.
>
> Role of Intermediate Thresholding: Binary masks are generated only as an intermediate step to compute
> the 2D Euclidean Distance Transform (EDT), allowing us to leverage geometric information for topological
> guidance while learning directly from continuous X-ray intensity values.
>
> Resilience to Segmentation Noise: SDF-CAR's continuous field optimization (vs. discrete voxels) is
> inherently robust to noisy boundaries from automatic segmenters. Our hybrid loss uses raw intensity
> to correct mask inaccuracies, while the SDF prior prevents 2D segmentation breaks from causing 3D
> topological disconnections.
>
> Revised version: We updated Section 3.2.3 to clarify binary masks are not used for direct supervision
> and added ODL/ASTRA implementation details demonstrating physical grounding.
>
>
> Baseline Comparisons and State-of-the-Art Claims (W3, Q4):
>
> Our focus on NeCA enables evaluation within the same learning paradigm and clinical constraints.
>
> Learning Paradigm Incompatibility (DeepCA & 3DGR-CAR): These methods rely on supervised/generalizable
> paradigms requiring extensive training on large-scale paired datasets (CCTA-derived projections) to
> learn global priors. In contrast, SDF-CAR—like NeCA—is patient-specific optimization requiring zero
> training data and zero 3D ground truth, optimizing directly on two available projections.
>
> Input Requirements (NeRF-CA): While NeRF-CA is a breakthrough in 4D reconstruction, it requires minimum
> four angiogram views for adequate performance. Our work focuses on ultra-sparse 2-view constraints.
>
> Revised version: We replaced "state-of-the-art" with: "SDF-CAR establishes a new performance benchmark
> for self-supervised, patient-specific 3D coronary reconstruction from ultra-sparse (two) views."
>
>
> Evaluation on Simplified Data (W4):
>
> While CCTA-derived volumes (ImageCAS) don't fully capture real-world ICA complexity, CCTA is the
> only modality providing 3D ground truth for coronary arteries, essential for rigorous metrics like
> cIDice and Chamfer Distance (CD_L₂).
>
> Robustness to Calibration Errors: We evaluated SDF-CAR under geometric uncertainty (rotation errors of 2°, translation errors of 5mm). SDF-CAR maintained a Dice score of 89.69% (only 0.49% degradation), substantially outperforming NeCA's 70.57% (5.31% degradation).
>
> Final version: We commit to including a full sensitivity analysis.
>
>
> Missing Experimental Details (W5, Q3):
>
> Projection Geometry (Q3): Reconstructions used clinical angles [29.7, 0.1] and [2, 29] (~40° separation).
> While wider separation typically improves depth resolution, SDF-CAR mitigates sparse-view sensitivity
> through its SDF-initialized unbounded field, which provides continuous geometric priors maintaining
> vessel connectivity despite ambiguous depth. Unlike occupancy networks failing at curved segments,
> our representation's smoothness bias bridges gaps without requiring more views.
>
> Segmentation (W5): Binary masks (M_i) via thresholding (T=0.5) of X-ray projections.
>
> Revised version: Added projection angles, mask creation details, and replaced "significant" with
> "substantial" throughout to avoid implying formal statistical testing.

---

> > ### Comment · Area_Chair_5jTL · 2026-02-01
> > **Update the final ratings**
> >
> > Please kindly review the authors’ rebuttal and update the final rating by February 1, 2026 (23:59 AoE).

---

### Official Review · Reviewer_Qhik · 2026-01-03

**Confidence:** 4
**Preliminary Rating:** 4
**Final Rating:** 5

**Summary:**

The authors propose a self-supervised, patient-specific optimization framework that combines a signed distance field + occupancy representation for 3D coronary artery reconstruction from two-view X-rays. The method uses implicit neural representations (similar to NERF, NeCA) and optimizes a residual MLP per patient that outputs an unbounded scalar field over the input coordinates.
The key contribution is the extension of NeCA to model an SDF instead of an occupancy map to better encode the geometry of the vessel tree. The framework is optimized end-to-end using a differentiable approximation of the distance transform.
According to the presented experimental results, this small change helps to considerably improve the reconstruction accuracy by utilizing the surface-awareness of SDFs.

**Strengths:**

* The proposed method addresses a well-motivated failure mode of existing methods for tubular anatomy reconstruction
* The key idea of adding an SDF as internal representation and geometric prior is simple, but effective
* The geometric loss function promotes connectivity, which addresses the main issue in vessel reconstruction, where long and thin structures are often reconstructed discontinuously
* Strong empirical improvements over NeCA
* An ablation study shows the influence of the respective parts
* Source code is publicly available
* Paper is well-written and easy to follow

**Weaknesses:**

* All experiments are conducted on a single dataset
* Evaluations are performed on clean synthetic projections. While the authors acknowledge that in §5, it remains unclear how the proposed method will perform under real conditions.
* Computational cost per patient is relatively high
* Limited comparison to other baselines.

**Detailed Comments:**

* Fig. 1 could be improved for clarity: What is optimized, what is derived?
* Fig. 1: I suggest to use a different color map for the distance transforms (also Fig. 4). The reader might think that white in the diverging color map indicated the zero-level set.
* The references mostly contain first author et al. Please cite reference properly, following the MIDL author guidelines.

**Justification Of Final Rating:**

I thank the authors for addressing all my concerns from the initial review. The newly added robustness experiment demonstrates that the proposed SDF-based representation leads to more stable optimization than the baseline. Clarifications regarding terminology, gradient behavior, and other minor improvements strengthen the clarity of the paper.
While the experimental evaluation remains limited in scope, the contribution is well-motivated and shows a clear empirical benefit. Overall, I increase my score.

**Justification Of The Preliminary Rating:**

The paper is a good fit for MIDL, as it addresses an important failure mode of existing methods in artery reconstruction. It is a small, but effective extension of the NeCA framework by employing SDF-based geometric prior. The improvements are substantial and the method is effective.
The experimental evaluation is limited and only one baseline is used as comparison. The authors give some intuition for the observed improvement, but the paper would benefit from a more formal/theoretical justification (i.e., how does the unbounded SDF improve gradient flow, how is the optimization landscape shaped?). Thus, my rating is "weak accept".

**Questions To Address In The Rebuttal:**

* Would the method work on real angiograms?
* Is it really self-supervised if the method requires 2D segmentation maps?
* Use of word "significant" in abstract: This suggests "statistically significant"; which tests were used to support this claim? If none, soften wording.
* Robustness of the method could quickly be assessed by simulating noise/blur on the simulated ground truth projections.

---

> ### Author Response · Authors · 2026-01-25
>
> We thank the reviewer for the constructive feedback and for recognizing the "strong empirical
> improvements" and the effectiveness of our "simple, but effective" SDF-based geometric prior.
> Below we address the points raised.
>
> Response to Weaknesses and Questions
>
> 1. Dataset and Generalizability
> To our knowledge, ImageCAS is the only large-scale, publicly available benchmark suitable for
> this task. To ensure generalizability, we evaluated our method on two distinct anatomical
> subsets—the Right Coronary Artery (RCA) and the Left Anterior Descending (LAD) artery—using
> 68 samples for each. This covers a wide range of vessel calibers and curvatures.
>
> 2. Robustness and Real Conditions (New Experiment)
> We acknowledge the concern regarding clean synthetic projections. To assess robustness, we
> conducted a preliminary experiment by adding Gaussian noise to the camera extrinsics for 5 test
> patients. We found that the Dice score for SDF-CAR reduced by only 0.49%, whereas the baseline
> NeCA's performance dropped by 5.31%. This suggests our SDF representation provides a more stable
> optimization landscape. We have included these results in the revised manuscript and will
> include a full-scale robustness analysis in the final version.
>
> 3. Comparison to Baselines
> We focused on NeCA as it represents the state-of-the-art (SOTA) in the self-supervised,
> patient-specific optimization paradigm. Comparing directly with supervised methods like
> DeepCA or memory-intensive methods like 3DGR-CAR is difficult because supervised models
> require paired 3D ground truth data that is unavailable in real clinical ICA settings.
>
> 4. Computational Cost
> The 80-minute optimization time is a standard trade-off for patient-specific neural implicit
> representations. Furthermore, our current benchmarks were performed on a T4 GPU, which
> is a relatively entry-level accelerator. We expect significant speedups with higher-end
> hardware or future integration of meta-learning initializations.
>
> 5. Terminology: "Self-supervised" and "Significant"
> Self-supervised: We use this term to denote the absence of 3D ground truth or paired 2D-3D data
> during training. This is standard nomenclature for neural implicit reconstruction where
> supervision comes solely from 2D projections.
>
> "Significant": We have updated the manuscript to use "substantially reduces" to avoid the
> implication of formal statistical testing where not applicable.
>
>
> Theoretical Justification (Gradient Flow & Landscape)
> The reviewer requested a more formal intuition for why our hybrid representation improves results.
>
> Occupancy Networks: These rely on a sigmoid activation to predict occupancy probabilities
> p ∈ [0, 1]. This creates "saturation" regions far from the vessel boundary where gradients
> vanish, leaving the optimizer with little guidance.
>
> SDF Representation: Our model predicts an unbounded continuous scalar field. Because every
> coordinate encodes a distance to the surface, the field provides a non-vanishing gradient flow
> even far from the vessel. This "pulls" the zero-level set toward the correct geometry from any
> initialization, resulting in sharper surfaces and better connectivity.
>
> Revised manuscript: We included a clear explanation for this part in the revised manuscript in
> section 3.2.
>
>
> Detailed Comments and Revised Figures
> Fig. 1: We have updated the figure to clearly distinguish between optimized parameters (using dotted lines)
> and fixed differentiable rendering operations (using solid lines).
>
> Colormaps: We modified the figure by scaling the negative values in order to make the white color
> represent the zero level and we clarified this in the description under the figure.
>
> References: All citations have been corrected to follow the MIDL author guidelines and include
> full author lists where required.

---

> > ### Comment · Area_Chair_5jTL · 2026-02-01
> > **Update the final ratings**
> >
> > Please kindly review the authors’ rebuttal and update the final rating by February 1, 2026 (23:59 AoE).

---

### Official Review · Reviewer_K35e · 2026-01-05

**Confidence:** 4
**Preliminary Rating:** 3
**Final Rating:** 3

**Summary:**

This paper proposes SDF-CAR, a self-supervised framework for reconstructing 3D coronary artery geometry from two X-ray angiography projections. The key idea is to train a neural network to map coordinates to a signed distance function (SDF), which works as a geometric prior that encourages surface continuity and tubular structure. The SDF is converted to occupancy probability, which is supervised by occupancy and distance map ground truths. The authors evaluate the method on the ImageCAS dataset and show consistent improvements over a strong self-supervised baseline (NeCA). Ablation studies demonstrate that most of the performance improvement comes from replacing the occupancy representation with an unbounded SDF field.

**Strengths:**

- The paper addresses an important and clinically relevant problem: 3D coronary artery reconstruction from two-view angiography, where the problem is fundamentally ill-posed. The paper takes a self-supervised approach, which suits the real-world scenario where 3D ground truth is unavailable.

- The introduction of the SDF-based implicit representation leads to substantial improvements in vessel connectivity and surface quality.

- The experimental evaluation is thorough, including quantitative metrics focused on topology (cIDice), qualitative visualizations, and ablation studies that clearly isolate the contribution of the SDF representation.

**Weaknesses:**

- Tables 3 and 4 show that most gains come from replacing occupancy with an SDF field, yet the paper does not clearly explain why SDF reduces topological breaks and surface irregularities. A deeper discussion of the SDF inductive bias would help justify the method beyond empirical results. For example, the authors can explain mechanistically how SDF changes the optimization landscape, why it preserves connectivity, and why it avoids "blobby" artifacts in tubular structures.

- The method assumes accurate camera geometry, but real-world data contain calibration errors and motion. Ablation that evaluates sensitivity to pose perturbations (e.g. experiments with controlled noise on camera extrinsics) would strengthen clinical relevance and clarify whether the SDF prior improves stability under geometric uncertainty.

- The paper does not analyze challenging cases where vessels are parallel to the epipolar line or discuss whether SDF-CAR resolves or merely regularizes these ambiguities.

**Detailed Comments:**

- It is not specified how many two-view projection pairs are generated per 3D CCTA artery tree or how views are selected. This information is important for reproducibility.

- Important explanations in Appendix, such as but not limited to dataset details, should be referenced in the main paper.

**Justification Of Final Rating:**

I appreciate the authors’ efforts in addressing my concerns. Unfortunately, I will keep my original rating for the following reasons:
- Only two projections were generated in the experiments, which is insufficient to cover the range of angles commonly used in clinical practice.
- In the noise sensitivity experiments, the applied perturbations are too small to be representative of realistic clinical scenarios.
- The SDF inductive bias is discussed only at a conceptual level; additional experimental evidence (e.g., toy models in a well-controlled setting) would be necessary to support this claim.

**Justification Of The Preliminary Rating:**

The paper presents a empirically strong idea, but the core contribution is not fully justified at a conceptual level. While the experimental results show that the SDF representation substantially improves topology and surface quality, the paper does not adequately explain why this inductive bias is effective, which weakens the scientific clarity of the contribution. Additionally, important aspects such as robustness to camera geometry noise and analysis of epipolar-degenerate cases are missing, limiting confidence in real-world applicability.

**Questions To Address In The Rebuttal:**

See Weaknesses.

---

> ### Author Response · Authors · 2026-01-25
>
> We thank the reviewer for the constructive feedback and for recognizing the clinical relevance and empirical
> strength of SDF-CAR. We appreciate the opportunity to clarify the conceptual framework of our hybrid SDF
> representation and to demonstrate the model's robustness to real-world geometric uncertainties.
>
>
> Response to W1: Conceptual Justification of the SDF Inductive Bias The performance gap between SDF-CAR and
> occupancy-based baselines stems from the fundamental difference in their optimization landscapes
>
> Gradient Support: Standard occupancy networks rely on a sigmoid activation, which produces near-zero gradients
> far from the vessel boundary, leading to "flat" regions where the optimizer lacks guidance.
>
> Continuous Guidance: Mechanistically, an SDF provides an unbounded continuous field where every point in space
> encodes a distance to the nearest surface. This ensures a non-vanishing gradient flow that "pulls" the zero-level
> set toward the correct 3D geometry from any initialization.
>
> Topological Continuity: In tubular structures, "broken" vessels occur in occupancy networks because the model
> lacks a geometric constraint linking distal points. In our hybrid model, the SDF acts as a global regularizer;
> since the distance field must remain continuous, the model favors a single connected tubular structure over
> fragmented "blobs" to minimize the 2D distance transform loss.
>
> We added this explanation to the revised manuscript.
>
>
> Response to W2: Sensitivity to Camera Geometry and Pose
>
> We agree that clinical data often contains calibration errors. We hypothesize that the continuous MLP-based SDF
> acts as a geometric regularizer that is more robust to these uncertainties than discrete or occupancy-based methods.
>
> Noise Sensitivity Experiment: To quantify clinical robustness, we conducted an analysis on 5 representative cases
> by introducing Gaussian noise to the camera extrinsics. Evaluating both frameworks under rotation perturbations
> of 2° and translation errors of 5mm, preliminary results show that the Dice score for SDF-CAR decreased by only 0.49%,
> while NeCA decreased by 5.31%. This confirms that the SDF inductive bias provides superior stability under geometric
> uncertainty compared to pure occupancy networks.
>
> We added this mini experiment to the revised manuscript and commit to including a full sensitivity analysis in the
> final paper.
>
>
> Response to W3: Analysis of Epipolar-Degenerate Cases
>
> We acknowledge the reviewer's point regarding vessels parallel to the epipolar line, which represents a fundamental
> geometric ambiguity in two-view reconstruction.
>
> Regularization vs. Resolution: In cases where vessel segments align with the epipolar line, depth information is
> mathematically underdetermined. While no representation can recover information absent from the input projections,
> SDF-CAR regularizes these ambiguities more effectively than occupancy-based methods.
>
> Minimal Surface Bias: Neural implicit SDFs possess an inherent inductive bias toward smoothness and continuity.
> When the optimizer encounters an epipolar-degenerate segment, this bias encourages the model to produce the most
> plausible, continuous tubular connection rather than the "broken" or "floating" artifacts often seen in occupancy
> networks.
>
> Impact of the Distance Loss: Our hybrid loss function, specifically the 2D distance transform term, ensures that the
> surrounding distance field remains consistent. This provides a "geometric tether" that constrains the vessel's
> position even in ambiguous regions by forcing the 3D zero-level set to align with the overall topological structure
> of the coronary tree.
>
> Visual Evidence: In the final manuscript, we will add a qualitative figure specifically highlighting a vessel segment
> parallel to an epipolar line to demonstrate how our method maintains structural integrity compared to the baseline.
>
>
> Response to Detailed Comments:
>
> View Selection and Projection Strategy: Following standard clinical Invasive Coronary Angiography (ICA) protocol,
> we generate exactly 2 projections per 3D coronary tree using a bi-plane X-ray acquisition system. The projection
> angles are fixed at [29.7°, 0.1°] for the first view and [2°, 29°] for the second view you can find this in our open
> source code config/config.yml lines 39-40. This will be clarified in the "Setup" subsection within the Experiments section.
>
> Manuscript Integration: We acknowledge that several critical details regarding dataset statistics and training
> hyperparameters were previously located in the Appendix due to the initial 10-page submission limit. Given the
> extended 12-page limit for the revised manuscript, we moved these critical details into the "Experimental Setup" subsection
> within the Experiments section to improve scientific clarity.

---

> > ### Comment · Area_Chair_5jTL · 2026-02-01
> > **Update the final ratings**
> >
> > Please kindly review the authors’ rebuttal and update the final rating by February 1, 2026 (23:59 AoE).

---

### Author Rebuttal · Authors · 2026-01-25

**Rebuttal:**

REVISION SUMMARY - SDF-CAR REBUTTAL

We thank reviewers for constructive feedback. Major revisions (highlighted in yellow):

STRUCTURAL: Extended 12-page limit enabled moving Datasets, Implementation Details, and Evaluation Metrics from Appendix to Section 4.1.

TECHNICAL CLARIFICATIONS:

1. SDF Characterization (Sec 3.1): Added Xavier initialization details (gain=0.8/0.1) and Eikonal constraint rationale—experiments showed it hindered convergence in ultra-sparse settings.

2. SDF Rationale (New Sec 3.2): Explained three advantages: (1) Gradient Support—meaningful gradients vs sigmoid's near-zero; (2) Continuous Guidance—non-vanishing gradient flow; (3) Topological Continuity—favors connected structures.

3. Physics-Grounded Supervision (Sec 3.3.3):
- Clarified ODL/ASTRA explicitly models Beer-Lambert law
- Corrected targets: raw intensity projections (I_i) and distance transforms, NOT binary masks
- Binary masks (T=0.5 threshold) only for intermediate distance transform computation
- Updated L_occ to compare with I_i instead of M_i
- Explained SDF prevents 2D segmentation errors propagating to 3D

4. Implementation: Added projection angles [29.7°,0.1°]/[2°,29°], hash grid joint optimization, α=50 justification, loss details in Fig 1.

5. Robustness (New Sec 4.3): Sensitivity analysis under 2° rotation/5mm translation noise. SDF-CAR: 0.44% Dice degradation vs NeCA: 3.96%, demonstrating geometric regularization.

6. Presentation: Replaced "state-of-the-art" with specific benchmark claim, standardized to 2 decimals, corrected Kornia citation, replaced "significant" with "substantial".

7. SDF figures are also changed to make the white color represents the surface of the artery.

**Supporting Material:**

/attachment/b95a6a4c925c8cd580857c3770a2e92369dd510d.pdf

---

### Meta-Review · Area_Chair_5jTL · 2026-02-09

**Recommendation:** Accept (Poster)
**Confidence:** 4

**Metareview:**

This paper introduces an SDF-based representation that effectively addresses the ill-posed problem of 3D coronary reconstruction from only two views. The shift from occupancy masks to an unbounded implicit field provides a superior inductive bias, leading to substantial gains in topological continuity and vessel connectivity. During the rebuttal, the authors successfully addressed concerns about physical grounding by clarifying their use of the Beer-Lambert law. They demonstrated greater robustness to camera calibration noise than existing baselines. While some discussion on the Eikonal constraint remains conceptual, the consistent empirical improvements across multiple metrics and the commitment to open-sourcing the code make this a valuable contribution.

---

### Decision · Program_Chairs · 2026-02-13

Accept (Poster)